# Current Optical Sensing Applications in Seeds Vigor Determination

**Jian Zhang** [1,2,*] , **Weikai Fang** [3], **Chidong Xu** [3], **Aisheng Xiong** [4] , **Michael Zhang** [5], **Randy Goebel** [5] and **Guangyu Bo** [3,*]

1 Faculty of Agronomy, Jilin Agricultural University, Changchun 130108, China
2 Department of Biology, University of Columbia Okanagan, Kelowna, BC V1V 1V7, Canada
3 Anhui Institute of Optics and Fine Mechanics, Chinese Academy of Sciences, Hefei 230031, China; weikaif@cmpt.ac.cn (W.F.); xcd@aiofm.ac.cn (C.X.)
4 Crop Genomics and Bioinformatics Center and National Key Laboratory of Crop Genetics and Germplasm Enhancement, College of Horticulture, Nanjing Agricultural University, Nanjing 210095, China; xiongaisheng@njau.edu.cn
5 XAI Lab, Department of Computing Science, Alberta Machine Intelligence Institute, University of Alberta, Edmonton, AB T6G 2E8, Canada; mlzhang@ualberta.ca (M.Z.); rgoebel@ualberta.ca (R.G.)
* Correspondence: jian.zhang@ubc.ca (J.Z.); ayli@cmpt.ac.cn (G.B.)

**Abstract:** Advances in optical sensing technology have led to new approaches to monitoring and determining crop seed vigor. In order to improve crop performance to secure reliable yield and food supply, calibrating seed vigor, purity, germination rate, and clarity is very critical to the future of the agriculture/horticulture industry. Traditional methods of seed vigor determination are lengthy in process, labor intensive, and sometimes inaccurate, which can lead to false yield prediction and faulty decision-making. Optical sensing technology offers rapid, accurate, and non-destructive calibration methods to help the industry develop accurate decisions for seed usage and agronomic evaluation. In this review, we hope to provide a summary of current research in the optical sensing technology used in seed vigor assessments.

**Keywords:** seed vigor; spectral detection technology; image detection technology; digital agriculture

## 1. Introduction

### 1.1. Definition of Seed Vigor

The quality of seeds is the key factor in accurately determining the quality and yield of crops. In a broad sense, seed quality is a comprehensive evaluation of single or multiple indicators such as vigor, purity, germination rate, clarity, and water content. Among the various evaluation indicators of seed quality, seed vigor can often comprehensively determine whether crop seeds can germinate normally, the degree of uniformity of emergence, and the ability of grown plants to resist diseases. Therefore, the overall term "seed vigor" is often used as a general indicator for evaluating seed quality [1]. The development of a scientific definition of seed vigor has always been closely related to the developmental level of agricultural technology and has gone through its own relatively long development stage. As early as 1950, the International Seed Testing Association (ISTA) proposed the concept of seedling vigor and initiated active research on "seed vigor". However, it was not until 1977 that ISTA formally defined the definition of seed vigor: "Seed vigor refers to the comprehensive performance of the activity intensity and seed characteristics of seeds or seeds during germination and emergence, and those that perform well are high vigor seeds, and those that perform poorly are low vigor seeds" [2]. This definition of seed vigor is essentially a comprehensive concept composed of multiple indicators such as germination rate, germination potential, field emergence rate, stress resistance, and plant hardiness.

Seed vitality is determined during the process of seed development. The basis for the formation of seed vitality is the measurement of the accumulation of a variety of

substances in the seed. With the maturity of the seed, the starch, protein, lipid, and other substances in the seed gradually accumulate, and the seed vitality increases until the seed is physiologically mature. At that point, the enzymes in the seed body are inactivated, the protein and nucleic acid are closely combined, the metabolism of the seed is continuously reduced, the embryo enters a quiescent state, and the seed obtains maximum vitality. With the typical modern practice of extending seed storage time, various metabolisms during seed storage can cause a series of changes, such as component changes, membrane structure damage, and loss of enzyme activity. The seeds enter a natural aging process, and the seeds lose their genetic integrity, thus reducing seed vigor [3].

Under a wide range of field conditions, seed vigor is the basis for sound performance crops and high yields and is the main indicator to measure field emergence and seed usage. The highly vigorous seeds typically show strong stress resistance, strong vitality, and storage resistance, which can significantly increase the germination rate of seeds, shorten their emergence time, and increase production and income. However, seeds with low vigor have poor stress resistance, long emergence time, and low seedling rate, which are likely to cause major losses to agricultural production. Therefore, in-depth research on the regulation mechanism of seed vigor, research and development of accurate and rapid seed vigor testing technology, and breeding of new varieties with high vigor are the only ways to promote the sustainable and healthy development of the seed industry.

### 1.2. Significance of Seed Viability Detection

The issue of seed vigor has attracted more attention from the agricultural fields of many countries across the world. This is particularly important because of the continuous growth of the global population, the continuous reduction of arable land, and the deteriorating agricultural production environment, especially in developing countries such as China and India. In China, the proportion of agricultural product yield reduction due to issues of seed vigor is between 10% and 20% per year [4]. The key to solving the above problems is to sow seeds with higher vigor on limited land resources and fulfill efficient agricultural production. Therefore, before the seeds are sown, accurate detection of seed vigor can improve the yield and quality of crops and help to achieve the objective of food production supply. It is obvious that accurate knowledge of seed vitality information is a necessary means to ensure agricultural production potential.

On the other hand, realizing the rapid identification and prediction of seed vigor will help to more deeply understand the different genetic mechanisms of high-vigor seeds, as well as the impact mechanisms of different maturity periods, storage conditions, planting conditions, and climatic conditions [5]. Improved understanding of the principles of deterioration, finding ways to improve seed vigor, and cultivating new varieties with high vigor will help realize these ideas. In production practice, it is beneficial to provide basic monitoring data in the research process of improved seed breeding, to prevent the circulation of uncertified and poor-quality seeds in the process of seed trade, and to guide seed companies to master effective storage management measures to improve seed processing and production efficiency. Confirming the vitality of this connection will improve the quality of the seeds.

In recent years, a lot of work has been done on seed vigor in the global agricultural field, and many countries have listed seed vigor as a key research and developmental topic of agricultural scientific research, which has become an active study in the field of digital agriculture. In China, colleges and research institutes such as the Institute of Botany, Chinese Academy of Sciences, Chinese Academy of Agricultural Sciences, China Agricultural University, and Jilin Agricultural University have also conducted significant research on seed vitality. To summarize, whether it is basic scientific research or the development of vigor improvements in the seed industry, there is an urgent need for high-precision and rapid detection of seed vigor. It is worth noting that the formation and development stages of seed vigor are not only affected by internal factors such as complex genetic regulation and metabolism but also easily affected by external factors such

as growth environment and storage environment. Therefore, non-destructive online or continuous detection of seed vigor has always been a technical bottleneck in the field.

### 1.3. Seed Vigor Detection Method

The International Seed Testing Association (ISTA) and Association of Official Seed Analysts (AOSA) have compiled the standard of "International Rules for Seed Testing" [6] and the Seed Vigor testing handbook [7], which includes at least eight kinds of seed viability determination methods such as conductivity measurement. This article mainly discusses the rapid detection methods and does not discuss the direct evaluation methods of seed vigor, such as germination assays. The non-destructive and rapid detection method of seed vitality mainly focuses on a variety of physical and chemical detection methods to measure the physical and chemical, image, spectral, and electrical characteristic parameters related to seed vitality and subsequently establishes the relationship between seed vitality and the measured characteristic parameters [8]. Related methods mainly include (1) detection methods based on the physical and chemical properties of seeds, (2) detection methods based on electrical properties [9], and (3) detection methods based on optical characteristics [10].

### 1.3.1. Seed Vitality Detection Technology Based on Physical and Chemical Characteristics

With the typical industry increase of seed aging time, the content of soluble protein and the activity of protective enzymes in seeds show an obvious decline. At this time, the content of hydrogen peroxide reductase, superoxide dismutase, superoxide dismutase, etc., in the seeds can be detected by conventional physical and chemical testing methods, such as tetrazolium staining, and the content of related enzymes in the seeds can be quantitatively analyzed. The detection results of physical and chemical methods have high precision and high reliability, but these methods require professionals to operate and apply special reagents, which is not conducive to rapid on-site monitoring of seed vigor but is rather considered as laboratory analysis.

### 1.3.2. Seed Vitality Detection Technology Based on Electrical Characteristics

Seeds are hygroscopic dielectrics. When seeds contain more water, their resistivity is relatively small, and the seeds can be measured as good conductors of electricity. When the seeds lose water and dry, they show higher resistivity, which diminishes conductivity. This research results show that the dielectric constant is negatively correlated with the vigor of seeds, and with the help of physical testing methods of the dielectric constant, the change of the dielectric constant can be used to identify the non-destructive detection of seed vigor. It should be noted that the measurement of conductivity takes more than 24 h and usually does not have an online detection capability on a production site. At the same time, the measurement of conductivity needs to follow a precise operation process, which is more suitable for laboratory analysis scenarios.

### 1.3.3. Seed Vitality Detection Technology Based on Optical Characteristics and Artificial Intelligence Methods

With the advancement of agricultural science and technology and the integration of multi-disciplines, emerging technologies are constantly being experimentally applied to the field of agricultural science and technology innovation. Many scholars regard the cross-integration of optics and biology as a foundation for new research for the non-destructive evaluation of seed vitality. Modern optical analysis technology is used to identify the material composition and appearance characteristics by analyzing the different reflections of the material in the visible light spectrum. For plant seeds, there are substances that determine their vigor, which can be manifested as different optical parameters. These differences can characterize the changes in the content of vigor substances during the process of seed vigor changes, and based on this, they can be measured. For grading of vitality, for example, Bi et al. used deep learning with machine vision to exploit the Swin Transformer to improve maize seed recognition [10]. From a large volume of observation

data obtained from sensing equipment (i.e., both camera and special optical devices), they develop spectral information processing and feature extraction methods, then extract those spectral and image characteristics that reflect the level of seed vitality. This provides the basis to establish an accurate mapping relationship between spectral and image features and seed vitality values. Note that improving the accuracy of seed viability prediction model analysis is the ultimate goal of seed viability detection data processing. In order to achieve these objectives, some advanced data and information processing methods are needed; for example, mature machine learning algorithms have been applied to hyperspectral data analysis in the construction of spectral feature analysis and vitality identification models of corn and habit seeds [10]. At the same time, deep learning is effective and popular for dealing with complex classification problems in large volumes of data, especially for seed viability feature construction models. Similarly, some scholars have used deep learning algorithms to analyze hyperspectral imaging data of cotton, loofah, and rice and constructed a discriminant analysis model for plant disease detection [11]. This work has produced stable and accurate vigor identification results. With the rapid development of computer technology, relevant algorithms have quickly emerged, so there will be more advanced methods developed and applied in the direction of seed vitality prediction models.

## 2. Progress in the Detection of Seed Vigor Based on Modern Optical Methods

Optics is an ancient science, but since the beginning of this century, with the advancement of laser technology, optical engineering, and information technology, optical detection technology has gained a resurgence. The theoretical basis for the application of modern optical methods to the detection of seed vigor is that light irradiation on the surface of seeds will produce reflection, scattering, and transmission phenomena, accompanied by changes in parameters such as frequency, phase, and polarization state. In different vigor stages, the contents and content of nucleic acid, starch, enzyme, and protein inside the seed have a predictable change, and the reflected, scattered, and transmitted light information generated by the interaction with the light field can inform the internal composition and quality of the seed.

Note that modern optical methods belong to the non-destructive detection method of seed vigor: optical methods can measure many parameters related to seed vigor. The changes in these parameters can indirectly reflect the status of seed vigor. The technical process for optical method detection research is to first collect the light field information of single or group seeds after the interaction with incident light, analyze the difference of light field information of seeds with different vigor, and then obtain accurate seed vigor through standard germination tests and other methods. By establishing a mapping between the light field information measures and seed vigor value, a seed vigor evaluation system is formed. Ultimately, this creates an overall evaluation system that can be used to predict seed vigor.

Because the traditional methods of seed vigor are based on physical, chemical, and electrical characteristics, the operation of detection equipment is complicated, the detection process is highly restricted by the external environment, and the detection accuracy is highly dependent on the operating experience of the personnel. In that context, the detection samples are easily damaged, and the detection period is long and time-consuming. The improvement trajectory of such methods can no longer meet the needs of the rapid development of the modern seed industry market. Therefore, vigor detection technologies must be developed in the direction of non-destructive, rapid, and high-precision [5]. Within this direction, optical detection methods do not require pretreatment of the seeds and, at the same time, can obtain the internal chemical information without destroying the seed. Therefore, it has unique technical advantages in the direction of accurate, non-destructive, fast, and even online detection of seed vitality. Existing optical detection methods for seed vitality mainly include machine vision detection, near-infrared spectroscopy, hyperspectral detection, Raman spectroscopy, fluorescence spectroscopy, and seed exhalation gas spectroscopy [12].

### 2.1. Machine Vision Method

Machine vision technology has advanced to integrate many research fields, such as image acquisition, image processing, and machine learning. During research on the detection of seed vigor, researchers have found that seed vigor is not only related to the genetic characteristics of the seeds but also closely related to the appearance and physical characteristics of the seeds, such as size, color, and texture. Machine vision technology working in the visible light band is essentially a detection method based on appearance features. It mainly simulates the human eye through image sensors such as charge-coupled imaging capture devices (CCDs), which collect images of seeds and then use computer technologies such as edge detection and feature extraction to simulate human vision. This provides for the extraction of physical information such as shape features, color features, and texture features of seeds, which can be combined with standard germination experiments of seed vigor. The combination of these techniques establishes the basis for a machine vision detection model of seed vigor to evaluate the status of seed vigor.

Since the 1990s, studies have launched in-depth research on the application of machine vision technology in agriculture. For example, McCormac et al. developed a lettuce seed vigor detection system based on machine vision technology [4]. The results of the study showed that the image analysis technique has the potential to determine the vigor of seeds. In China, scholars have successively developed similar methods for pepper seeds [5], which exploits a machine vision detection system for the vigor of perilla, wheat, and other seeds. The accuracy rate of the detection results is usually between 80 and 90%, and this system provides the basis for an online detection application for vegetable seed vigor [13].

Traditional machine vision technology mostly works in the visible light band. With the advancement of infrared devices, infrared thermal imaging technology has significantly advanced. As an extension of traditional machine vision technology, infrared methods can be applied to the field of non-destructive detection of seed vitality. Because of the influence of water absorption, respiration, and other metabolic activities of seeds during seed germination, the surface temperature will exhibit small changes which infrared thermal imaging technology can quantify. These temperature changes are closely related to seed vitality [14], thus distinguishing the seed differences in temporal vitality intervals [15].

Analyzing the existing research literature, the current problems in the application of machine vision technology in the field of seed vigor detection focus mainly on the fact that, when assessing seed vigor indicators, current machine vision methods extract vigor-related parameters from only the physical parameters of the seed surface features. However, the volume of indicative information is relatively small, and there is room for improvement. Generally speaking, with the advantage of inexpensive key equipment components and relatively mature algorithms, machine vision technology has significant promise for further development.

### 2.2. Infrared Spectroscopy Technology

With the help of the interaction between light and matter, spectroscopy can extract the internal vigor components and content information of seeds, so it has greater application potential in the detection of seed vigor. Among various spectroscopic techniques, near-infrared spectroscopy (NIRS) has been extensively studied. NIRS belongs to the class of imaging techniques known as molecular absorption spectroscopy. The principle is that specific functional groups in different molecules produce specific absorption indicators at specific wavelengths, which can be exhibited as absorption spectra in broadband light sources. As mentioned above, during the storage of seeds, the internal protein, starch, fat, and cellulose components will change, which will cause changes in seed vigor. The above substances are rich in hydrogen-containing groups such as C-H (aliphatic hydrocarbon), C-H (aromatic hydrocarbon), C=O (carboxyl group), OH (hydroxyl group), and N-H (amino group). Because of abundant absorption properties, one can identify obvious differences in absorption wavelength and intensity [16]. For example, Jin et al. extracted the difference

in near-infrared spectra between viable rice seeds and inactive seeds [17], which more intuitively reflects the above characteristics; the results are shown in Figure 1.

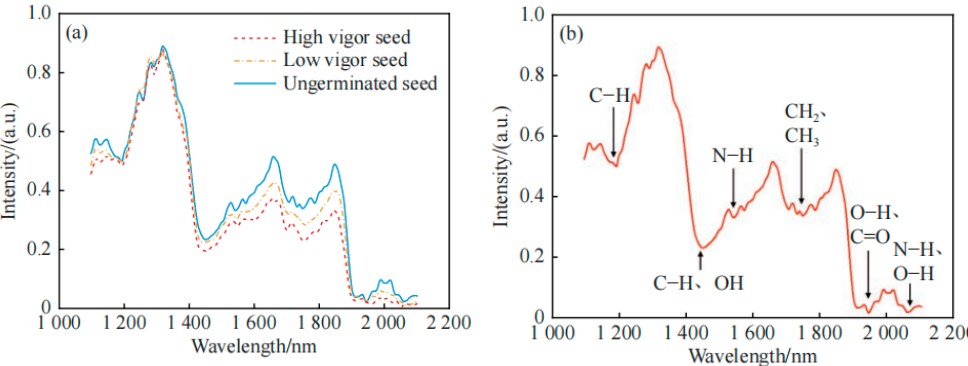

**Figure 1.** Near-infrared absorption spectrum of rice seeds; (**a**) The average curves of transmission spectra of rice seeds with different vigor characteristics; (**b**) Near-infrared absorption spectrum of high vigor seeds.

As shown in Figure 1, the spectra of viable and non-viable seeds have significant peak/valley differences in the bands of 1150–1220 nm, 1410–1450 nm, 1510–1540 nm, 1660–1800 nm, 1910–1950 nm, and 1980–2100 nm. The valleys represent groups such as amino N-H, hydroxyl O-H, and carbonyl C=O, respectively (Figure 1b). It can be seen that the changes in protein, starch, fat, and other components during the aging process of seeds can be characterized by this difference of infrared absorption spectrum, so as to extract identification characteristics of seed vitality.

Infrared spectroscopy, as a rapid non-destructive testing method, has previously attracted the attention of scholars earlier. For example, Tigabu et al. first applied near-infrared spectroscopy analysis technology to the detection of seed vigor. Their experiment obtained pine seeds with different vigor parameters that varied through accelerated aging. With the help of infrared spectroscopy technology, they successfully distinguished aged and unaged pine seeds. The accuracy rates of pine seed identification at the same level reached 80%, 90%, and 75%, respectively [8]. Al-Amery et al. used near-infrared spectroscopy to measure the standard germination rate and seed vigor of soybean seeds. By establishing a partial least squares prediction model, the seed vigor grades were classified, and two kinds of soybean seeds with low vigor and high vigor were identified. Note that the corrected rates were 80%~100% and 96.3%~96.6%, respectively. This is also the first non-destructive prediction model of near-infrared spectrum soybean seed vigor provided by the academic field [9]. NIRS technique has been reported for its application on seed non-destructive evaluation with pepper [12], watermelon [13], and tomato [14]. These studies helped to establish the viability of NIRS models of seed quality and have confirmed their application for seed quality evaluation.

Compared with the traditional infrared spectroscopy technology, the Fourier transform infrared spectroscopy technology that has recently emerged provides more powerful spectral analysis capabilities. For example, Larios et al. used Fourier transform infrared spectroscopy to study the vigor of soybean seeds and established a recognition model using modeling methods such as support vector machines. Their results showed that, under laboratory conditions, the correct rate of classification and identification of soybean seeds with different vigor could reach 100% [11].

In October 2006, the first near-infrared spectroscopy conference was held in China. This meeting became a milestone in the history of China's near-infrared technology development. Since then, a number of excellent research reports on the application of infrared spectroscopy to seed vigor have emerged. For example, Ling et al. used near-infrared supercontinuum laser spectroscopy to study three kinds of rice with different seed vigor, and the results showed that the accuracy of the prediction results reached more than

95% [18]. Fan et al. used near-infrared spectroscopy [19], studied the vigor of wheat seeds, and the results showed that the prediction accuracy of the model was not less than 84%, indicating that the near-infrared spectroscopy technology has the potential to predict the vigor of wheat seeds. Overall, near-infrared spectroscopy technology is widely used in wheat, rice, corn, soybean, and other crops, fir, Masson pine, Slash pine, Loblolly pine, and other forest seeds, as well as tomatoes, in the study of vigor detection of vegetable seeds [20,21].

Summarizing the existing problems in the application of infrared spectroscopy technology to the detection of seed vitality has identified the following aspects. On the one hand, infrared spectroscopy technology is susceptible to interference from working environment factors, and the extraction of spectral characteristics often needs to consider the noise caused by factors such as temperature and moisture interference. The correction of relevant environmental factors thus needs to be considered, which results in a relatively large modeling workload. On the other hand, the material components in the seeds are rich and complex, and interference factors such as the overlap and superposition of near-infrared spectral absorption peaks have always existed, which also affected the quantitative detection ability of infrared spectroscopy to a certain extent.

### 2.3. Hyperspectral Imaging Technology

As mentioned in Sections 2.1 and 2.2, machine vision technology and spectroscopy technology have their own outstanding advantages. How to realize the combination of the advantages of the two has naturally become the goal pursued by the developers of optical analysis instruments. Thanks to the successful development of high-performance imaging spectrometers, hyperspectral imaging (HSI) has become a reality. Each pixel of HSI can obtain a piece of multi-spectral data, which can reflect the material composition information at a certain position on the target surface; by combining a large number of pixels, seed image information can be reconstructed to achieve the effect of map integration. HSI technology can effectively integrate image technology and spectral technology, increase the dimension of information detection, and thus can analyze seed sample information more comprehensively [22].

Hyperspectral imaging technology was first applied in the field of geographic information remote sensing and gradually applied to the field of seed vigor detection in the past ten years. For example, Baek et al. used hyperspectral imaging technology to quickly and non-destructively distinguish the viability of soybean seeds, and the test accuracy was close to 100% [23]. Ambrose et al. used hyperspectral imaging technology in the 400–2500 nm band to identify two corn seeds with different vigor levels, and the recognition accuracy of the established models reached 97.6% and 95.6%, respectively [24]. Perez et al. used hyperspectral imaging technology to study the seed vigor of the Japanese juniper, and the results showed that hyperspectral imaging technology could be used to effectively predict the vigor of the Japanese juniper [25].

In China, relevant scholars have also carried out a lot of research on the application of hyperspectral imaging technology to the detection of seed vitality. The research targets include crops such as rice, wheat, soybean, and corn. Some progress has also been made in the construction of discriminant models. Pang et al. used hyperspectral imaging technology to study four kinds of corn seeds with different vigor and constructed a seed vigor identification model by using various modeling methods such as a support vector machine and extreme learning machine. The results showed that the identification of different vigor seeds was accurate. The rate reached over 90% [26]. Zhang et al. used hyperspectral imaging technology to detect wheat seed vigor and established a seed vigor detection model using spectral data sets at different positions on the seed surface. The results showed that the established vigor detection model had high accuracy [27]. Li et al. detected the vigor of rice seeds based on hyperspectral imaging technology, and the results showed that the accuracy of classification and identification of seeds with different vigor could reach 94.44% [28].

As analyzed above, compared with machine vision technology and near-infrared spectroscopy technology, hyperspectral imaging technology has the ability to extract the characteristics and internal components of samples at the same time. Therefore, hyperspectral imaging technology has gradually become an active direction for seed vigor detection. At the same time, the analysis of the existing research also found that hyperspectral imaging research generally has a series of problems, such as high spectral data dimensionality, data redundancy and interference information, and complex construction of seed vitality mapping models, which still need to be improved by domestic and foreign scholars.

### 2.4. Tunable Diode Laser Absorption Spectroscopy

Spectral technology can not only directly detect seed components but also approach the challenge of seed vitality detection by detecting gas components produced during seed metabolism. Respiration is a comprehensive indicator of plant seed metabolism. Most seeds need to complete life activities through respiration, such as cell division and differentiation during early seed germination. Scholars have carried out a lot of research on the correlation between seed respiration and seed vigor, and Woodstock et al. discovered there is a significant positive correlation between seed respiration rate and germination rate [29]. Kalpana et al. [30] found the respiration of seeds showed a positive correlation between strength and vigor, and the above research conclusions become the theoretical basis for judging seed vigor by using the components of seed respiration.

In the presence of oxygen, seed respiration manifests as aerobic respiration, which consumes $O_2$ and releases $CO_2$. Under anaerobic conditions, seeds decompose by storing glucose themselves to produce ethanol and $CO_2$. Therefore, by detecting the change in $CO_2$ gas concentration produced by seed respiration, the seed vitality can be judged through the information on seed respiration intensity. Scholars at home and abroad have tried to use different gas sensors to detect the $CO_2$ produced in the process of seed respiration and judge the vigor of seeds. Among various $CO_2$ sensors, Tunable Diode Laser Absorption Spectroscopy (TDLAS) has better detection accuracy and lower limit than electrochemical or semiconductor sensors and can accurately detect the $CO_2$ concentration produced by seed respiration, and thus has been favored by researchers. The principle of TDLAS technology exploits a distributed feedback laser (Distributed Feedback Laser, DFB) by adjusting the DFB operating temperature and the size of the driving current and adjusting the wavelength of the laser output laser so that the laser is "selectively" absorbed by the gas to be measured. Macroscopically, the laser intensity is weakened, and the concentration of the gas to be measured can be reversed by using the change of the laser intensity. TDLAS technology has excellent single gas molecule selectivity and can achieve online response; combined with optical absorption cell technology and wavelength modulation technology, its detection limit can reach hundreds of ppt levels.

TDLAS technology has been mainly used in industrial field detection, atmospheric greenhouse gas monitoring, and other fields, while there are relatively few research reports on seed breath detection in the agricultural field. However, in recent years, scholars at home and abroad have strengthened research work in related fields. In China, Jia and others used TDLAS technology [31], and the $CO_2$ released during the respiration process of corn seeds was detected; combined with the results of the standard seed vitality germination experiment, the results showed that the average value of the correlation coefficient between the $CO_2$ concentration produced by respiration and the vitality index was 0.975, and the respiration intensity strongly correlated with the vitality index. These experimental results also show that, while limited by the existing device level of TDLAS technology, it is currently difficult to accurately calculate the respiration intensity value of seeds under high dynamic resolution. This affects research progress of the detection of seed respiration intensity, which indicates the need for future research.

Note that current research on seed respiration and seed vigor is mainly qualitative, and a large number of seed respiration detection and seed germination experiments still need to be carried out to quantitatively study the correlation between seed respiration

intensity and seed vigor. On the other hand, although existing studies have shown that the respiration intensity of seeds such as corn, rice, and Chinese fir has a certain positive correlation with their germination rates, there are still some scientific issues in the process that have not been clearly understood, such as in the same genetic strain. Specific issues, such as the quantitative relationship between seed respiration intensity and seed vigor and the quantitative relationship between seed vigor and respiration intensity among different genetic lines, still need to be continuously studied.

### 2.5. Other Optical Detection Technologies

Fluorescence detection technology, Raman detection technology, photoacoustic spectroscopy detection technology, and other advanced optical detection methods have also been investigated for the development of non-destructive detection of seed vitality.

The application of chlorophyll fluorescence detection technology in the field of seed viability detection has also received attention in recent years. Anhui Institute of Optics and Fine Mechanics, CAS, has studied fluorescence detection techniques for years and developed a system of material composition three-dimensional platform. It has been used in digital agriculture research (Figure 2).

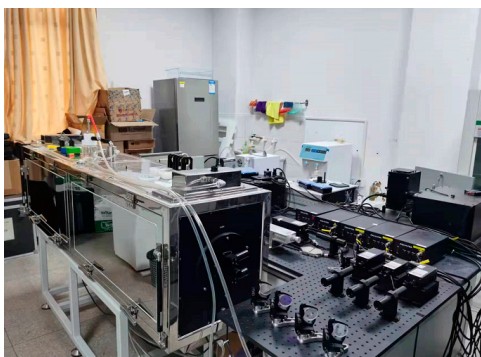

**Figure 2.** Material composition three-dimensional platform.

When the visible or near-infrared wavelength beam irradiates the seed epidermis, the chlorophyll on the seed coat will release energy in the form of fluorescence. By measuring the fluorescence spectrum released by the chlorophyll on the seed coat, relevant information on the seed vitality level can be obtained. It is confirmed that the smaller the peak signal of the fluorescence spectrum of the seed epidermis is, the lower the chlorophyll content is, and the higher the vigor of the seed is. Because this method is only specific to chlorophyll, it can reduce the influence of other optical radiation noise on the signal, so this method has certain technical advantages. Jalink and others first adopted fluorescence detection technology [32], studied the vigor of cabbage (*Brassica oleracea* L.) seeds, and confirmed that the chlorophyll fluorescence signal was negatively correlated with seed vigor. The analysis of existing research reports also found that the chlorophyll fluorescence detection method still needs to solve the technical difficulties, such as poor seed vigor grading effect due to the influence of chlorophyll content differences in seed samples.

Raman spectroscopy is a scattering spectrum technique that can provide detailed information about molecular vibrations. Since Raman peaks are usually clear and emerge in a relatively narrow band (only a few wavenumbers), it is useful in the feature identification and analysis of organic macromolecules. Raman methods have unique technical advantages and have formed the basis for a standard Raman analysis test instrument. Anhui Institute of Optics and Fine Mechanics, CAS, has carried out research on Raman Spectroscopy detection techniques. A handhold Raman Spectroscopy detection device which can be used for seed Raman spectroscopy detection is shown in Figure 3.

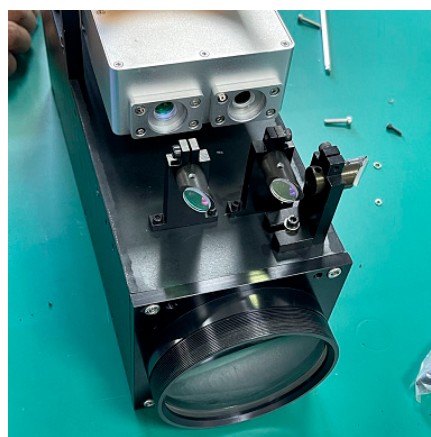

**Figure 3.** A handhold Raman Spectroscopy detection device.

As Raman spectroscopy shows great potential for rapid analysis, its related work in the field of general agricultural detection is also developing rapidly. More typically, Ambrose et al. used a Raman spectrometer (1700–3200 cm$^{-1}$) to identify viable and inactive corn seeds [33], with an accuracy rate above 93%. The analysis of existing research reports also found that the Raman scattering signal is weak, which plagues its application in the field of seed viability detection, and related problems still need breakthroughs in key devices.

Photoacoustic spectroscopy technology (Photoacoustic spectrometry technology, or PAS) is a detection technology based on a photoacoustic effect. Its detection principle is that the light source irradiates and heats the target. The ambient air is induced to vibrate weakly and generate sound waves, and the photoacoustic spectrum of the target can be obtained by detecting the weak sound wave signals. Different from traditional absorption spectroscopy, PAS does not directly detect the spectrum based on photon characteristics but detects the periodic heat flow in the non-radiative excitation phenomenon after the sample is illuminated, which changes the laser modulation frequency to adjust the light transmission. This measurement reveals the depth of the seed so that it can detect the material composition of the sample at different depths. Pardo et al. used photoacoustic spectroscopy technology [34] and carried out vegetable seed vitality detection research; their test results show that the light absorption of unaged seeds is higher, and the results show that the scheme of predicting seed vigor by photoacoustic spectroscopy is feasible. In general, photoacoustic spectroscopy is still a relatively new cutting-edge technology, and it is worth investing more resources in research. The current research developments of seed vigor detection procedures based on modern optical methods are assembled in Table 1.

**Table 1.** Current research developments of seed vigor detection procedures using modern optical methods.

| Detection Method | Main Principle | Working Light Band | Core Device | Typical Application |
|---|---|---|---|---|
| Machine Vision Technology | Optical Imaging of Seeds | Visible light/near-infrared | CCD Imaging device | Perilla, Wheat, etc. [4,5,13] |
| Infrared Spectroscopy | Infrared Absorption Spectrum of Organic Matter | Near-infrared | Infrared spectrometer | Pine seeds, soybeans, wheat, rice, corn, etc. [8,21,35–37] |
| Hyperspectral Imaging Technology | Reflection/Absorption Spectrum of Seed Surface | UV to NIR | Broadband light source + imaging spectrometer | Pepper, Rice, Avocado etc. [17,28,38,39] |
| Turnble Laser Absorption Spectroscopy | Infrared Absorption Spectrum of Organic Matter | Near-infrared to mid-infrared | Tunable laser + lock-in amplifier Tunable laser + lock-in amplifier | Maize [29], pigeonpea [30] |

**Table 1.** *Cont.*

| Detection Method | Main Principle | Working Light Band | Core Device | Typical Application |
|---|---|---|---|---|
| Raman spectrometer | Raman Frequency Shift Spectroscopy of Organic Compounds | UV to Visible Light | Pulse laser + Raman spectrometer | Maize [10] |
| Photoacoustic spectroscopy | Photoacoustic Effect Excited by Matter Without Radiation | UV/Visible | Laser + weak acoustic wave detector | Lettuce, leaf beets, etc. [33] |
| Fluorescence spectroscopy | Fluorescent Effect of Organic Matter | UV to visible Light | Fluorescence Spectrometer | Cabbage and other vegetable seeds [12] |

## 3. Development Trends of Key Common Optical Technologies

In theory, optical detection technology has powerful detection capabilities and extremely high measurement accuracy. However, in practice, it is found that the detection capability of the system is often limited by the band response range and wavelength resolution capability of the device; the detection accuracy of the system is often limited by factors such as the optical and electrical noise of the device. On the other hand, even for seeds of the same species, seeds from different planting batches or from different origins will inevitably have differences in chemical composition and phenotypic characteristics, which will affect the spectrum and image of the seeds, creating a large workload for evaluation model construction. The problems faced by optical detection of seed vigor are more complicated. The key to solving this problem lies in whether optical methods can effectively support research progress that combines key devices, modern information technology, and the correlation method of optics and chemometrics.

### 3.1. Development Status of Key Components

The key components of spectrum or image analysis equipment are concentrated in key units such as the laser light source, the imaging device, and the spectrum analysis device.

Traditional spectral analysis methods usually use a halogen lamp to synthesize a broadband light source, and after irradiating the seed target to be detected, a reflection, scattering, or transmission spectrum is generated. That seed image is received to achieve the purpose of extracting the apparent characteristics and composition information of the seed. Usually, the shell of the seed is thick, and the performance of the halogen light source is limited in terms of intensity, directionality, and stability. This produces a weak interaction between the light field and the seed, which restricts the detection capability of the equipment. The emergence of laser light sources has significantly improved the intensity, directionality, and stability of the light source. However, the common commercial lasers on the market are mostly single-wavelength laser sources with narrow linewidths or lasers with limited wavelength tuning capabilities near the output wavelength (no more than tens of nm), and these lasers are often only able to monitor a single material component within the seed. To meet the simultaneous detection of multiple components of the seed, the wavelength coverage of the light source is usually required to reach 400 nm~2500 nm.

Therefore, whether to increase the bandwidth of the light source while ensuring the emission power of the laser light source and then obtain a spectral signal with sharp features and a high signal-to-noise ratio or an image with high color and signal-to-noise ratio has become the key to improving the ability of optical analysis of seed vitality.

In the past 10 years, there have been breakthroughs with new optical crystal material technology, including the development of nonlinear optical technology and fiber laser technology. As a consequence, the research on supercontinuum light sources in the world has achieved rapid development, especially the infrared supercontinuum laser source, which is the key to spectral analysis. One of the most exciting new technologies to emerge in the field is the so-called Supercontinuum light source technology, which uses semiconductor lasers to pump photonic crystal fibers. These devices can generate broadband spectra covering

400~2500 nm, and the highest spectral power density can reach milliwatts/nanometers [40]. At the same time, these have the spectral range of traditional halogen lamps and monochromatic lasers. Directionality and high energy characteristics, strong light source penetration, and wide spectrum range, while satisfying the simultaneous detection of multiple material components, can also extract data from deep into the seeds and can obtain the full-band spectrum and images of the changes in the internal components of the seeds as much as possible. This kind of information can further improve the performance of optical detection of seed vitality.

While supercontinuum laser technology has made major breakthroughs in the past ten years, high-tuning precision DFB lasers for the detection of seed exhalation breath intensity [41] and quantum cascade lasers for the detection of macromolecular substances in the mid- and far-infrared bands have both achieved substantial progress. The development of this new type of light source has also made new breakthroughs in the optical detection of seed vigor and laid the foundation for key devices.

The spectral analysis component is a necessary foundational component of modern optical analysis methods. Its main function is to perform phenotype discrimination and component analysis on substances by obtaining the spectrum after the interaction between light and substances. The main technical route of common spectral analysis components is to use discrete optical devices such as gratings and prisms to separate light fields of different frequencies in space, and then use area arrays of CCD imaging devices for measurement, and finally achieve the purpose of spectral imaging measurement. The working mechanism of discrete device combination often leads to a series of problems, such as the high cost of spectral analysis components, poor integration ability, and low long-term working stability. This means the measurement requirements of high precision, high consistency, and on-chip integration of equipment are difficult to achieve. Therefore, the development of new spectral analysis components to achieve wider detection bands, finer wavelength resolution, and miniaturized on-chip integration capabilities has become an urgent breakthrough direction.

### 3.2. Modern Information Technology and Spectral or Image Preprocessing Algorithms

Affected by their own metabolism and storage environment, after the seed reaches its peak vitality, its vitality gradually decreases, and the change process of the vitality in a short period of time is subtle. The key parameters are usually only evident in the content of some key enzymes, hormones, and other vital substances have changed, and such changes are reflected in differences in spectral or phenotypic image information of seed vigor, often very subtle (e.g., often a few percent change in spectral intensity). On the other hand, restricted by the technical level of existing devices, the information richness of spectral and image data that can be obtained at present is limited. With the complexity of these two factors, the development of seed viability evaluation equipment has remained a significant challenge.

When combined with modern information technology, carrying out spectrum or image preprocessing research is an effective means to alleviate the lack of capability of the above-mentioned detection devices. However, the current conventional spectral preprocessing methods, such as the derivative method to extract information bursts and other algorithms, lack the ability to effectively amplify the differences in optical characteristics of different vigor seeds. Therefore, in order to achieve accurate identification of seed vigor, it is necessary to propose a more reasonable spectral preprocessing algorithm for small differences in optical characteristics and to amplify and preprocess the differences in spectral or image information so as to effectively reflect the small changes in seed vigor. The above spectral or image-preprocessing technologies often rely on the development of modern information technology.

### 3.3. Correlative Methods of Optics and Chemometrics

Optical detection technology is an indirect analysis method, so it is necessary to establish an effective "spectrum-vitality" evaluation model as a bridge between the spectrum and seed vigor information. However, the existing optical detection of seed vigor often directly correlates the measurement results of optical information with standard seed germination experiments. The "spectrum-vitality" evaluation model established by these methods fails to expose the required "spectrum-omics-seed" insight. The complete research chain of vitality makes the mechanism of the model unclear, and the accuracy is limited. However, the chemometric analysis methods to further obtain the material components in the process of seed vigor changes can be combined with an exploration of more refined differences in material components of different vigor seeds. This coupling will provide a material basis for determining seed vigor so that a more quantitative evaluation model can be established. It can be seen that strengthening the research on the correlation method between modern optical technology and chemometrics is also a key potential breakthrough to improve the accuracy of optical detection technology [42].

## 4. Conclusions and Perspective Discussion

Due to the wide variety of seed types and traits, traditional seed viability detection methods have difficulty meeting the needs of rapid and non-destructive detection of seed vigor. Optical technology can provide different vigor detection methods in a targeted manner according to different vigor expressions of seeds. For example, for seeds with a strong correlation between the vitality index and the degree of browning of the seeds, machine vision technology provides the basis for differential assessment; for seeds whose components have undergone drastic changes after deterioration, near-infrared spectroscopy or hyperspectral technology can be used to judge the vigor of the seeds; and for seeds that need to be detected in batches, TDLAS technology can be used to detect their respiration intensity and evaluate the vitality of the seed population, for some seeds that require extraction of parameters that are revealed only by looking into the depth of seeds. This provides the basis for detecting seed vitality components, and photoacoustic spectroscopy is an effective method to detect this component of seed vitality.

With the development of key devices and detection methods, optical technology has received extensive attention in the field of assessing seed viability. However, the analysis of current research reports is still based on laboratory research. There is still a lack of commercial seed viability optical detection equipment in the market. The relevant problems are mainly reflected in the following three aspects: (1) Although the non-destructive detection method of seed vigor has the advantage of being fast and non-destructive, it is restricted by factors such as the development level of key photoelectric detection devices and many problems still need to be overcome in the current research stage. For example, machine vision technology can judge the vigor of seeds through the information of seed phenotype, but not all kinds of seed vigor are related to the physical information of seed phenotype; near-infrared spectroscopy and hyperspectral technology can obtain the internal composition spectrum information of seeds, therefore, spectral components contain a lot of redundant information and noise, and breakthroughs in image noise reduction and fast image processing technology are urgently needed. (2) The current optical detection methods for seed vigor require a standard germination test for each batch or variety of seeds, followed by subsequent correlation analysis with the data obtained through optical technology detection to obtain a vitality evaluation model. In the process of model construction, the focus is on the mapping relationship between optical features and the apparent features of seed vigor, but there is a lack of an effective correlation process between chemometrics and optical features. In addition, there is relatively little research on the mechanism of seed vigor, and it is impossible to establish an effective relationship mapping. The quantitative relationship between optical features and viability grades affects the validity of evaluation models. (3) The cost of laser light sources and precision optical devices involved in optical detection methods is generally high, especially the

high price of key devices for hyperspectral imaging detection, which limits its commercial application prospects.

In view of the above problems, the research direction that needs to be strengthened for the modern optical detection of seed vigor includes: (1) Strengthen the theoretical research on the correlation between optical characteristics and seed components through in-depth research on the optical characteristics, omics, and vigor of seeds, obtaining a clear mechanism related to optical properties and seed vigor, and finally establish a refined and more adaptable seed vigor detection model. (2) Due to the different advantages of optical detection technologies with different principles, it is often difficult to fully obtain seed vigor information with a single technology, which affects detection accuracy. It is undoubtedly one of the future research directions to explore the fusion of multi-sensing technologies, realize multi-method detection and multi-parameter combination, and improve the effectiveness of crop seed vigor detection. On the other hand, although hyperspectral imaging and other detection methods are powerful, they are limited by the cost of devices, and their commercial application is currently underdeveloped. The combination of low-cost infrared absorption spectroscopy technology and machine vision technology can also increase the dimension of information and achieve the purpose of further improving the accuracy of seed vitality detection, thus having better commercial application prospects. (3) Strengthen engineering application research and promote the early industrialization of detection technology research. Internationally, giants such as Monsanto Company of the United States have carried out application attempts of online seed viability detection devices, which have initially demonstrated the commercial value of the equipment.

In order to further promote the application of optical seed viability detection technology, the field of engineering research needs to comprehensively consider the three elements of performance, cost, and reliability and select the technology with high commercial feasibility for engineering. On the other hand, with the development of electronic information and automation technology, it is necessary to strengthen the combination of detection technology and intelligent classification devices, develop integrated detection and sorting equipment, and promote the commercial application of related equipment as soon as possible.

**Author Contributions:** Conceptualization: J.Z. and G.B.; software, G.B.; validation, C.X., W.F. and A.X. resources, G.B. and J.Z.; data curation, C.X. and G.B.; writing—original draft preparation, J.Z. and G.B.; writing—review and editing, J.Z., M.Z. and R.G.; visualization, G.B.; supervision, J.Z. and G.B.; project administration, C.X.; funding acquisition, A.X., G.B., C.X. and J.Z. All authors have read and agreed to the published version of the manuscript.

**Funding:** Jilin Agricultural University high-level researcher grant (JLAUHLRG20102006).

**Data Availability Statement:** Not applicable.

**Acknowledgments:** Authors would like to thank Jianyu Lu for the reference checking for this manuscript.

**Conflicts of Interest:** The authors declare no conflict of interest.

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
