# Peer review of "Current Optical Sensing Applications in Seeds Vigor Determination"

_agronomy, doi:10.3390/agronomy13041167_

Round 1

Reviewer 1 Report

Dear Authors,

you have presented an interesting review paper, in which you juxtaposed the current state of knowledge regarding traditional methods of seed improvement-improving their vigor-with modern methods of optical analysis. 

Please address a few points, which I have included below:

- please provide information on whether any crop species have distinctive low seed vigor (genetically determined), and which have high?

- how is it standard in field production to assess seed vigor? what experiments are generally conducted?

-can seed vigor be predicted? what variables would you then include in such a model?

- please complete the DOI references of publications (where possible) in the references. 

Author Response

Re: agronomy-2223924

Dear reviewer

Thank you so much for your time and effort reviewing our manuscript, agronomy-2223924 entitled “Current Optical Sensing Applications in Seeds Vigor Determination”.

Based on your suggestions and commons, we have made revision with our manuscript. Our response to your points is highlighted here as well.

- please provide information on whether any crop species have distinctive low seed vigor (genetically determined), and which have high?

Thank you for the suggestion.  Crops like Cannabis sativa. L have relatively low seed vigor.

- how is it standard in field production to assess seed vigor? what experiments are generally conducted?

Seed providers would conduct seed germination test before their sale to growers.  Sometime, seed vigor can various due to environmental and location conditions

-can seed vigor be predicted? what variables would you then include in such a model?

Efforts are made to predict seed vigor from breeding and agronomical consideration.  

- please complete the DOI references of publications (where possible) in the references. 

Thanks for the kind reminding.  DOI reference of articles are checked and added.

Reviewer 2 Report

The article focuses on an interesting theme. Undoubtedly, optical sensing and AI technologies are the future in this field. The English language presents few typos, nothing that compromises the article's understanding.  The article is well organized. I do understand that, as the research area is new, it is hard to find references applied to agriculture. But, for a review article, from my point of view, it is necessary to add AI techniques and more information for the readers about optical sensing. So, in order to overcome the lack of articles on optical sensing, I would like to suggest the authors add more photos and details concerning the equipment and hardware necessary to carry out experiments in this field. For the AI part, it is necessary to map the main AI techniques used and compare them (maybe showing the main machine learning / deep learning algorithms in a table?). I am confident that following these suggestions, the authors will have a great contribution for the field. 

Author Response

Re: agronomy-2223924 

Dear reviewer

Thank you so much for your time and effort reviewing our manuscript, agronomy-2223924 entitled “Current Optical Sensing Applications in Seeds Vigor Determination”. We appreciate your encouragement with it.

Based on your suggestions and commons, we have made revision with our manuscript. Our response to your points is highlighted here as well.

The article focuses on an interesting theme. Undoubtedly, optical sensing and AI technologies are the future in this field. The English language presents few typos, nothing that compromises the article's understanding.  The article is well organized. I do understand that, as the research area is new, it is hard to find references applied to agriculture. But, for a review article, from my point of view, it is necessary to add AI techniques and more information for the readers about optical sensing. So, in order to overcome the lack of articles on optical sensing, I would like to suggest the authors add more photos and details concerning the equipment and hardware necessary to carry out experiments in this field. For the AI part, it is necessary to map the main AI techniques used and compare them (maybe showing the main machine learning / deep learning algorithms in a table?). I am confident that following these suggestions, the authors will have a great contribution for the field. 

Thank you so much for the commons. We have added machine learning/ deep learning algorithms as you suggested in revised version.

Reviewer 3 Report

Dear co-authors:

Overall, the manuscript presents valuable information regarding the use of optical remote sensing for monitoring and determining a crop seed vigor. However, before accepting publication the authors should consider some comments as listed below:

1- I recommend to use English editing service. Some sentence uncomplete and also type mistakes (for exp: line 222, 218, 150 and 161). 

2- Could you analysis based on review of literature how many studies used optical remote sensing data and conventional methods? Adding some tables for comparing different methods for evaluation the seed vigor and its viability.

3- Double check References and adding a link (doi) to each article to help the reader to access the article. 

Kind Regards, 

Reviewer 

Author Response

Re: agronomy-2223924 

Dear Reviewer, 

Thank you so much for your time and effort reviewing our manuscript, agronomy-2223924 entitled “Current Optical Sensing Applications in Seeds Vigor Determination”. We appreciate your encouragement with it.

Based on your suggestions and commons, we have made revision with our manuscript. Our response to your points is highlighted here as well.

1- I recommend to use English editing service. Some sentence uncomplete and also type mistakes (for exp: line 222, 218, 150 and 161). 

Thank you for suggestion and your critiques. We have made editing with all the typo and change.

2- Could you analysis based on review of literature how many studies used optical remote sensing data and conventional methods? Adding some tables for comparing different methods for evaluation the seed vigor and its viability.

Thank you so much for suggestion.  We would like to continue follow the research and report in our subsequent reports.

3- Double check References and adding a link (doi) to each article to help the reader to access the article. 

Thank you for suggestion and your critiques, We have made corrections.

Round 2

Reviewer 1 Report

I accept.

Author Response

Thank you so much much for your kind consideration and encouragement. 

Reviewer 3 Report

Dear Authors:

Thanks for your efforts in the updating version. 

Kind regards,  

Author Response

Dear Reviewer,

Thank you of your kind critique and helpful suggestions. Your effort for sure made our manuscript improved. 

Best Regards,

Jian